# Genome-Resolved Co-Infection by *Aeromonas veronii* and *Shewanella* sp. in Koi Carp: A Zoonotic Risk for Aquarists

**DOI:** 10.3390/microorganisms14010036

**Published:** 2025-12-23

**Authors:** Gorkem Tasci, Nihed Ajmi, Soner Altun, Izzet Burcin Saticioglu, Muhammed Duman

**Affiliations:** Department of Aquatic Animal Diseases, Faculty of Veterinary Medicine, Bursa Uludag University, 16059 Bursa, Turkey; grkmrocker12@gmail.com (G.T.); nihed.ajmi.95@gmail.com (N.A.); saltun@uludag.edu.tr (S.A.); iburcinsat@gmail.com (I.B.S.)

**Keywords:** metabolic pathway, aquarium fish disease, co-infection, fish disease, one health

## Abstract

Co-infections are increasingly recognized as drivers of disease in ornamental fish, yet their genomic underpinnings and zoonotic implications remain underexplored compared to farmed species. Leveraging a One Health perspective, we investigated an acute mortality event in koi carp and characterized a co-infection by opportunistic aquatic bacteria that are also implicated in human disease. We isolated *Aeromonas veronii* and *Shewanella* sp. from a moribund koi using culture, biochemical assays, and MALDI-TOF MS, then generated draft genomes and performed orthology (OrthoVenn3), pathway annotation (KEGG BlastKOALA/Mapper), secondary-metabolite mining (antiSMASH), and virulence/resistome screening (VFDB/CARD), complemented by antimicrobial susceptibility testing. Clinically, affected fish showed dropsy/ascites, scale loss, abnormal buoyancy, and reduced activity. Phylogenomics positioned *A. veronii* Koi-2 within the *A. veronii* complex near species thresholds (ANI ~96.1%; dDDH ~70.2%), while *Shewanella* sp. Koi-1 formed a distinct lineage below accepted cut-offs relative to *S. seohaensis* (ANI ~95.9%; dDDH ~67.6%). The virulome comprised 194 loci in *A. veronii* Koi-2 and 152 in *Shewanella* sp. Koi-1 is enriched for adhesion, secretion, iron uptake, and immune-evasion functions. Genotype–phenotype agreement was high for multidrug resistance: *Shewanella* sp. encoded OXA-436 and *rsm*A, matching β-lactam resistance and reduced fluoroquinolone/phenicol susceptibility, whereas *A. veronii* carried *tet*(A), OXA-1157, *cph*A3, *sul*1, and *aad*A3 consistent with tetracycline, β-lactam, sulfonamide, and aminoglycoside resistance profiles. In conclusion, genome-resolved diagnostics confirmed a mixed *Aeromonas–Shewanella* co-infection with broad virulence potential and convergent resistance mechanisms, supporting the routine use of genomics to distinguish single- versus mixed-agent disease and to guide dual-coverage, mechanism-aware therapy in ornamental fish medicine while informing zoonotic risk mitigation.

## 1. Introduction

The global ornamental fish trade is a multi-billion-dollar industry, with the annual legal trade value estimated to range between USD 15 and 30 billion [1,2]. Despite its growing popularity, less than 1% of this trade is attributed to the public aquarium sector [3]. Koi carp (*Cyprinus carpio* var. *koi*) are among the most popular freshwater ornamental fish species, both as pets and as display animals in public aquaria, and as of 2024, they represent a substantial market share with an estimated trade value of USD 2.7 billion [4,5]. When kept in large ponds or aquaria, their striking color patterns and graceful swimming behavior are reported to evoke a sense of tranquility that sustains hobbyists’ interest while creating significant commercial opportunities for breeders [6,7].

Single or mixed infections frequently trigger disease outbreaks in fish under various stress conditions, leading to significant economic losses for aquaculture producers [8]. Concurrent infections can substantially impact fish health, altering the progression and severity of numerous diseases [9]. Co-infections observed in fish emerge from complex interactions shaped by genetic relatedness among pathogens, environmental factors and pre-existing diseases. These interactions are critical in determining pathogen transmissibility, virulence levels, and host survival outcomes [10]. In this context, many bacterial pathogens have been identified as causative disease agents in koi carp. These include species from the genera *Shewanella*, *Aeromonas Vibrio*, *Bacillus*, *Citrobacter*, *Edwardsiella*, *Flavobacterium*, *Mycobacterium*, *Klebsiella*, *Proteus*, *Providencia* and *Serratia* [8,11,12,13,14]. Establishing definitive etiology in ornamental fish mortality events is inherently challenging. Multiple bacterial, viral, fungal, and parasitic agents may co-occur; clinical signs are often nonspecific, and diagnostic workups may be constrained by financial, logistical, or welfare considerations. As a result, many reports can only infer the relative contribution of individual agents based on patterns of isolation, pathology and supporting genomic or phenotypic evidence rather than exhaustive exclusion of all possible pathogens.

The genus *Aeromonas* comprises Gram-negative bacteria that are widely distributed in aquatic habitats, among aquatic organisms, and in food products. They are clinically important as opportunistic pathogens in fish and humans [15,16]. *A. veronii* is a significant opportunistic pathogen in aquatic systems and is widely distributed in nature, exhibiting remarkable environmental adaptability [17]. In fish, it is the primary cause of epizootic ulcerative syndrome and hemorrhagic septicemia [18,19,20]. While *Aeromonas salmonicida* remains the best-known *Aeromonas* species in finfish pathology, recent studies have highlighted that *A. veronii* and related taxa can also cause severe systemic infections in a wide range of hosts, including ornamental fishes [14]. In recent years, the frequency of large-scale *A. veronii* outbreaks has increased, leading to serious losses in aquaculture; the bacterium can also infect amphibians and warm-blooded animals, posing a threat to food security [21,22,23]. It has been associated with sepsis, gastroenteritis, and other infections in humans, particularly in the elderly and children with reduced immune function [24,25,26,27]. Furthermore, infections have been documented even in individuals with intact immune function. [28,29]. *Shewanella* is another bacterial genus capable of causing disease in koi and is present in both freshwater and marine ecosystems, representing a latent threat to aquaculture [30]. Various species have been isolated from healthy farmed and wild ornamental fish, as well as frequently from diseased fish [31,32,33,34]. Typical disease manifestations include ulcerative and necrotic skin lesions [33,35]. However, in some conditions, the infection can be characterized by darkening in color, lethargy, erosive lesions on the mouth, pale gills, lens atrophy and exophthalmos [31,32,33,34]. Visceral findings include hemorrhage—especially in the kidney and spleen—ascites, hepatic pallor, and petechial hemorrhages in the swim bladder [31,33,35]. In addition, several *Shewanella* species have shown zoonotic potential, having been linked to human septicemia, skin and soft-tissue infections, biliary tract infections, peritonitis, and emphysema [36,37,38,39].

This study presents the genome-based characterization of *Aeromonas veronii* and *Shewanella *sp. concurrently isolated from an adult koi carp that presented with marked ascites, abnormal buoyancy and reduced activity. The data obtained provide novel insights into the potential virulence traits of these opportunistic pathogens and their possible interactions during co-infection. Moreover, the findings contribute to a more comprehensive assessment of waterborne zoonotic risks in line with the ‘One Health’ approach.

## 2. Materials and Methods

### 2.1. Sampling and Bacteriological Identification

In 2024, an acute mortality event involving large koi carp was reported in an ornamental system in Bursa, Turkey. According to the submitting aquarist, multiple koi in the affected aquarium reportedly died with similar clinical signs, including swimming imbalance, abdominal distension and anorexia. One moribund adult koi that exhibited the most pronounced and representative lesions was submitted to our laboratory for a detailed diagnostic work-up and to resolve antimicrobial treatment ineffectiveness. A necropsy was performed in accordance with international fish disease diagnostic protocols and animal welfare principles [40]. For microbiology, aseptic swabs/tissue samples were collected from the liver, kidney, spleen, heart, and abdominal fluid and streaked onto Tryptic Soy Agar (TSA; Merck, Cat. No. 105458). Plates were incubated at 22 °C for 24–48 h. All macroscopically distinct colony types were initially subcultured, and two dominant morphotypes were retained for downstream analyses. For long-term preservation, isolates were suspended in liquid medium containing 15% (*v*/*v*) glycerol and stored at −80 °C. Colony morphology was recorded on TSA. Gram staining was performed using a commercial kit (bioMérieux, Marcy l’Étoile, France). Oxidase activity was assessed with Bactident Oxidase strips (Merck, Darmstadt, Germany; Cat. No. 1.00181), catalase activity with 3% hydrogen peroxide, and motility in SIM Medium (Thermo Fisher Scientific, Waltham, MA, USA). Additional biochemical traits were determined with the API 20NE rapid identification system (bioMérieux, Marcy l’Étoile, France).

### 2.2. MALDI-TOF MS Pre-Identification

Presumptive identification of selected isolates was performed by MALDI-TOF MS (matrix-assisted laser desorption/ionization time-of-flight mass spectrometry; Bruker Scientific LLC, Billerica, MA, USA) following the manufacturer’s instructions. From 24 h cultures, a single colony was smeared as a thin film onto a 96-spot steel target using a sterile loop. Within 30 min, 1 µL of 70% formic acid was applied to each spot and allowed to air-dry at room temperature. Subsequently, within 30 min, 1 µL of HCCA matrix solution (α-cyano-4-hydroxycinnamic acid dissolved in 50% acetonitrile, 47.5% deionized water, and 2.5% trifluoroacetic acid) was added and dried. Targets were loaded for acquisition, and spectra were processed and matched in BioTyper v3.1 (Bruker Daltonics, Bremen, Germany). In accordance with the vendor’s interpretive criteria, scores ≥ 2.0 were considered reliable at the species level.

### 2.3. Genome-Based Identification

Genomic DNA was extracted from each isolate using the NucleoSpin Microbial DNA kit (NucleoGene, Istanbul, Turkey) per the manufacturer’s protocol. DNA concentrations were measured on a Qubit 4™ fluorometer with the dsDNA HS assay kit (Thermo Fisher Scientific, Waltham, MA, USA). Long-read libraries were prepared from 400 ng high–molecular–weight DNA using the Oxford Nanopore ligation sequencing kit (SQK-NBD114-24; Oxford Nanopore Technologies, Oxford, UK) without fragmentation, following the PromethION protocol. Libraries were loaded onto a PromethION flow cell and run on a P2 Solo device (ONT) for 24 h. High-quality reads were assembled de novo with Unicycler v0.4.6 (hybrid pipeline) [41]. Draft assemblies were submitted to GenBank, and contigs > 1000 bp were annotated using the NCBI Prokaryotic Genome Annotation Pipeline (PGAP) [42].

For comprehensive phylogenomic and digital DNA–DNA hybridization analyses, genome sequences were compared to type-strain genomes using the Type Strain Genome Server (TYGS; https://tygs.dsmz.de/, accessed 23 December 2024) [43]. Lineage placement and comparative phylogeny were additionally inferred with the Bacterial Genome Tree Service (BV-BRC; https://www.bv-brc.org/app/PhylogeneticTree, accessed 15 August 2025), including reference genomes of *Shewanella* spp. and *A. veronii* from human, aquatic, and terrestrial sources obtained from GenBank. Following Olson et al. [42], the BV-BRC workflow was used to build a codon-aware phylogenomic tree from 1000 single-copy genes. Tree visualization and annotation, including host and geographic metadata overlays, were performed in Interactive Tree of Life (iTOL; https://itol.embl.de/).

### 2.4. Functional Genome Analyses

Antimicrobial resistance and virulence genes were screened using the Resistance Gene Identifier (RGI) within the Comprehensive Antibiotic Resistance Database (CARD; https://card.mcmaster.ca/analyze/rgi, accessed 13 August 2025) and the Virulence Factor Database (VFDB; http://www.mgc.ac.cn/cgi-bin/VFs/v5/main.cgi, accessed 17 August 2025) [44,45,46]. Because VFDB does not provide a curated *Shewanella*-specific virulence gene set, the virulence-associated loci in the *Shewanella* sp. KOI-1 draft genome was additionally queried by BLASTn (https://www.ncbi.nlm.nih.gov/home/genomes/, accessed 30 October 2025) (NCBI) against known *Shewanella* virulence regions deposited in NCBI; regions meeting the thresholds of ≥70% nucleotide identity and ≥80% coverage were recorded as putative virulence genes. Genome annotation was performed using Prokka v1.14.6, accessed via Proksee (https://proksee.ca/, accessed 30 October 2025), with default parameters [47]. Prophage regions were predicted with PHASTEST (https://phastest.ca/, accessed 20 October 2025), classifying scores > 90 as “intact,” 70–90 as “questionable,” and <70 as “incomplete” [48]. Potential human pathogenicity was evaluated with PathogenFinder (https://cge.food.dtu.dk/services/PathogenFinder/, accessed 17 August 2025) [49]. Ecological distribution and habitat association were inferred with Protologger (https://protologger.bi.denbi.de/, accessed 27 September 2025) [50].

### 2.5. Genome-Level Inference of Bacterial Interactions

Annotated proteomes of both isolates were retrieved from GenBank and subjected to orthologous clustering in OrthoVenn3 (https://orthovenn2.bioinfotoolkits.net, accessed 30 September 2025) [51]. The analysis delineated shared and isolate-specific protein clusters between the two taxa. For functional annotation, each proteome was analyzed in KEGG BlastKOALA (https://www.kegg.jp/blastkoala/, accessed 10 October 2025) to obtain KEGG Orthology (KO) identifiers, which were organized per species and explored with KEGG Mapper (https://www.genome.jp/kegg/mapper/, accessed 10 October 2025) to enable pathway-level, side-by-side comparisons [52,53,54]. Secondary-metabolite biosynthetic capacity was profiled with antiSMASH v7.0 (https://antismash.secondarymetabolites.org, accessed 12 October 2025); predicted biosynthetic gene clusters (BGCs) were recorded with their class, genomic coordinates, similarity scores, and nearest MIBiG references [55]. antiSMASH outputs were examined alongside KEGG pathway assignments to infer putative metabolic and functional points of interaction between the isolates.

### 2.6. Antimicrobial Susceptibility Testing

Antibiotic susceptibility was assessed by disk diffusion following CLSI VET03/VET04-S2 guidance for bacteria from aquatic animals [56]. Test agents included tetracycline (30 µg), ampicillin (10 µg), gentamicin (10 µg), ciprofloxacin (5 µg), florfenicol (30 µg), doxycycline (30 µg), oxytetracycline (30 µg), flumequine (30 µg), amoxicillin (25 µg), erythromycin (15 µg), enrofloxacin (5 µg), and trimethoprim/sulfamethoxazole (1.25/23.75 µg) were provided from Oxoid (Thermo Fisher Scientific, USA). Bacterial suspensions standardized to 0.5 McFarland were inoculated onto Mueller–Hinton agar (MHA; Merck, Cat. No. 103872), antibiotic disks were applied, and plates were incubated at 28 °C for 24–28 h. Inhibition-zone diameters were measured in millimeters; isolates exhibiting no measurable zone (0 mm) were recorded as resistant. *Aeromonas salmonicida* subsp. *salmonicida* ATCC 33658 served as the quality-control strain, with zone diameters verified against CLSI VET03/VET04-S2 criteria. Interpretation of susceptibility results took into account known intrinsic resistance patterns in *Aeromonas* and *Shewanella* spp., and only deviations from expected intrinsic profiles were considered indicative of acquired resistance.

## 3. Results

### 3.1. Necropsy Findings and Phenotypic Characterization of Isolates

Macroscopic examination revealed widespread scale loss and surface hemorrhages with marked abdominal distension (Figure 1A). Visceral inspection showed advanced ascites and multifocal hemorrhages, most prominent in the heart (Figure 1B,C). The swim bladder displayed abnormal morphology with nodular deformation (Figure 1D). Bacteriology of the heart and anterior kidney yielded two dominant, mucoid, glossy colony morphotypes: a light-cream colony (designated KOI-1) and a dark-brown colony (KOI-2), both purified to single strains. Both isolates were motile, oxidase-positive, and strongly catalase-positive. Microscopy showed Gram-negative rods.

MALDI-TOF MS pre-identification yielded high-confidence matches to *Aeromonas veronii* and *Shewanella* sp., with score values ≥ 2.0, supporting the phenotypic and genomic assignments.

### 3.2. Genome Analysis

The *Shewanella* sp. Koi-1 and *A. veronii* Koi-2 strains have been deposited in the NCBI GenBank database. The Koi-1 isolate consists of three contigs, with a genome size of 4.85 Mb and a GC content of 47.96%. In contrast, the Koi-2 isolate comprises two contigs, exhibiting a slightly larger genome size of 4.89 Mb and a more compact, GC-rich genomic profile with a GC content of 58.33%. Genome annotation revealed a total of 4255 and 4614 predicted genes in Koi-1 and Koi-2, respectively, of which 4119 (*Shewanella* sp. Koi-1) and 4455 (*A. veronii* Koi-2) were identified as protein-coding sequences (CDSs). Both isolates contained a high number of rRNA (10–11 copies), tRNA (103–123), and ncRNA regions, indicating substantial translational capacity. Furthermore, 245 pseudogenes were identified in the *A. veronii* Koi-2 genome, compared to 84 in the *Shewanella* sp. Koi-1 genome, indicating that these two species have undergone distinct evolutionary adaptations.

Phylogenetic analysis demonstrated that the *Shewanella* sp. Koi-1 isolate is closely related to *S. seohaensis*, *S. xiamenensis*, and *S. mangrovisoli*, but is positioned on a separate branch, indicating a distinct phylogenetic lineage. This suggests that Koi-1 and *S. seohaensis* may share a common evolutionary ancestor. However, the Koi-1 isolate represents a genetically distinct phylogenetic clade (Figure 2, Appendix A). The average nucleotide identity (ANI) values calculated between Koi-1 and *S. seohaensis* CCUG 60900 were ANIb: 95.9% and ANIm: 96.26%, while the digital DNA–DNA hybridization (dDDH) value was determined to be 67.6%. As this value falls below the 70% threshold generally used for species delineation, it supports the classification of *Shewanella* sp. Koi-1 is a distinct species despite its phylogenetic proximity to *S. seohaensis*. Furthermore, the isolation of *Shewanella* strains from aquatic, fish, and human sources in the same geographical region as Koi-1 lends further support to the hypothesis that this strain may be capable of occupying diverse ecological niches and potentially engaging in zoonotic interactions.

In the phylogenetic tree, all isolates were identified as belonging to the *A. veronii* species. The isolates were color-coded based on their origins: blue for aquatic sources, green for human sources, and brown for terrestrial sources (Figure 3, Appendix A). The Koi-2 isolate is indicated in blue with white text, representing its morphological characteristics, and is clustered among the aquatic-origin isolates. Phylogenetically, the Koi-2 isolate is positioned within the *A. veronii* species complex, displaying a high degree of similarity and close genetic relatedness to other aquatic isolates. The average nucleotide identity (ANI) values between Koi-2 and the reference strain *A. veronii* CECT 4257 were calculated as follows: ANIb = 96.13%, ANIm = 96.45%. The digital DNA–DNA hybridization (dDDH) value was determined to be 70.2%, which is very close to the 70% threshold commonly accepted for species delineation. These findings indicate that the Koi-2 isolate shares a high degree of genetic similarity with *A. veronii* and is most likely positioned within this species (Figure 3).

Furthermore, the presence of both human-, aquatic-, and terrestrial-origin isolates within the same phylogenetic cluster supports the notion that *A. veronii* is a broadly distributed, opportunistic, and potentially zoonotic bacterial species.

### 3.3. Functional Genome Analysis

Functional genome analysis of *Shewanella* sp. strain KOI-1 showed a single circular chromosome (~4.6–4.8 Mbp; Figure 4) densely populated with coding sequences on both strands and featuring rRNA operons (e.g., 23S rRNA), numerous tRNA loci, a tmRNA, and CRISPR elements. Annotated genes included core information-processing functions (*rpoB*, *rpoC*, *dnaE*, *infB*, *leuS*, *rne*, *recB*/*recC*, *mfd*, *lon*), chromosome organization (*smc*), and central metabolism (*mdh*, *gdhB*, *gltB*, *purL*, *narG*, *fdnG*). Cell-envelope and transport modules were represented by *lptD* and outer-membrane nutrient receptors (e.g., *btuB*), while environmental sensing/regulation featured multiple *rcsC* paralogs. The map also contained loci for secretion/interaction systems (e.g., *hcp*), alongside many distributed hypothetical proteins (Figure 4).

Functional genome analysis of *A. veronii* strain KOI-2 revealed a circular chromosome of ~4.6–4.7 Mbp (Figure 5), densely populated with coding sequences on both strands and annotated features including rRNA operons (e.g., 23S rRNA), multiple tRNA loci, a tmRNA, and putative CRISPR elements. Gene annotations encompassed core information-processing functions (e.g., *rpoC*, *uvrA*, *recC*, *mfd*), cell division/chromosome maintenance *(smc*, *mukB)*, and central metabolism (e.g., *metH*, *carB*, *gltB*, *gdh*, *fumB*, *putA*, *purL*). Cell envelope and transport-related loci were prominent, including outer-membrane biogenesis/secretion genes *(bamA*, *tamB)*, type II secretion component *(xpsD)*, and multiple efflux/transport systems *(macB*, *mexB*, *cusA*, *emrD)*. Environmental sensing and regulation were represented by several two-component and transcriptional regulators *(rcsC* paralogs, *torS*, *arcA)*, alongside motility/chemotaxis and surface structures (e.g., *flhA*). Numerous hypothetical proteins were distributed across the genome, interspersed with the above functional categories (Figure 5).

The prophage analysis revealed the presence of distinct prophage regions within the genomes of *A. veronii* strain KOI 2 and *Shewanella* sp. strain KOI 1 (Table 1). *A. veronii* strain KOI 2 harbored two prophage regions: a 38.6 kb intact prophage associated with PHAGE_Escher_500465_1 (NC_049342), encoding 54 proteins and exhibiting a GC content of 58.74%, and a 38.3 kb questionable prophage region linked to PHAGE_Entero_mEp237 (NC_019704), comprising 58 proteins with a GC content of 55.76%. In comparison, *Shewanella* sp. strain KOI 1 contained a single 45.7 kb questionable prophage region associated with PHAGE_Entero_cdtI (NC_009514), encoding 50 proteins and exhibiting a GC content of 47.94% (Table 1).

### 3.4. Pathogenic Potential

PathogenFinder classified the *Shewanella* sp. Koi-1 genome as a human pathogen with a probability of 0.478. Input proteome coverage was 0.76%. The analysis reported 4227 sequences totaling 1,388,782 bp (longest sequence 4323 bp; shortest 30 bp; average segment length 328.0 bp). Fourteen pathogenic gene families and 18 non-pathogenic families were matched. Representative hits included TonB-dependent receptor proteins with high amino-acid identity to *Shewanella baltica* genomes (e.g., 95.19% to CP000753, protein AB509755; 92.02% to CP001252, protein ACK48044) and a Na^+^ symporter (92.73% identity to CP000753, protein ABS80828). A conserved hypothetical protein also matched a *Serratia marcescens* genome (94.77% identity to AE014299, protein AAN58732).

PathogenFinder classified the *A. veronii* Koi-2 genome as a human pathogen with a probability of 0.519. Input proteome coverage was 0.56%. The analysis comprised 4610 sequences totaling 1,411,921 bp (longest sequence 5359 bp; shortest 30 bp; average segment length 306.0 bp). Twelve pathogenic gene families and 14 non-pathogenic families were matched. Representative hits included a signal peptide peptidase showing 92.51% identity to *A. salmonicida* A449 (protein ABO98081), a group II decarboxylase with 95.04% identity to *A. hydrophila* ATCC 7966 (protein ABK36070), and a protein from *Photobacterium profundum* SS9 with 94.19% identity.

### 3.5. Virulence and Virulence-Associated Genes

Virulence gene analysis indicated that both isolates possess substantial pathogenic potential. In total, 194 virulence-associated genes were identified in the *A. veronii* Koi-2 isolate and 152 in the *Shewanella* sp. Koi-1 isolate. These genes predominantly correspond to functions related to adhesion, secretion systems, iron acquisition, and immune evasion.

In *Shewanella* sp. Koi-1 strain, proteolysis was represented by a Zn^2+^ metalloprotease (M48 family; *zmp1/M48*), consistent with extracellular protease activity. Lipopolysaccharide (LPS) biogenesis and export were encoded by a complete *lpt* module (*lptB/C/D/E/F/G*) and core lipid A pathway genes (*lpxA/B/C/D/K/L/M*, *waaA*, *kdsA*, *rfaE/hldE*) alongside the ABC flippase *msbA*. Fatty-acid synthesis (FAS-II) components (*fabA/B/D/F/G/H/Z*, *fabV*, and regulator *fabR*) were also present. LOS/LPS modification genes included *eptA*, *gmhA*, and an *ArnT* homolog. Oxidative-stress defense comprised *katB* and *katG*. Cysteine/glutathione acquisition and redox buffering were supported by *cysE/K/M*, *cydC/D*, *gshA/B*. Biotin metabolism genes (*birA*, *bioA/B/C/D/F*, *accB*) were encoded. Exopolysaccharide and capsule formation were indicated by *galU*, *ugd*, *galE*, *glmM*, *pgi*, and capsule loci (GfcC family; Wzi). Multidrug efflux capacity was encoded by the RND pump *acrA/acrB* with the outer-membrane channel *tolC*. Respiratory versatility linked to virulence ecology included nitrate-sensing/reduction genes (*narQ*, *napA/B*). A complete type II secretion system (T2SS/Gsp) was present (*gspC/D/E/F/G/H/I/J/K/L*). Quorum-sensing capacity was indicated by *luxS* (AI-2 pathway). Iron acquisition and heme utilization machinery included heme biosynthesis (*hemA–H*), ferrous-iron uptake (*feoA/B*), and cytochrome *c* maturation (*ccmA–I*). Motility and chemotaxis systems were encoded by core chemosensory regulators (*cheA/B/R/V/W/Y/Z*), flagellar structural genes for hook/basal-body assembly (*flgA–M*, *flgN/O*), the export/assembly apparatus (*flhA/B/F/G*; *fliE/F/G/H/I/J/K/L/M/N/O/P/R*), and motor proteins (*MotA/MotB*, *PomA*). Surface adhesion and biofilm determinants encompassed type IV pili (*pilB/M/N/P/T/V/W/Z/Q* and related components), MSHA pili (*mshI/J/K/F/P/L*), and curli fibers (*csgB/D/E/F/G*) (Appendix A).

For *A. veronii* Koi-2 strain, cytotoxic and enterotoxic factors included pore-forming aerolysin/hemolysin family toxins (e.g., *aer*A and *hly*A) together with cytotonic/cytotoxic enterotoxins (*alt*, *ast*, *act*). Type III secretion–linked effector activity was represented by ADP-ribosylating toxin *aexU*. Secretory capacity was supported by complete type II secretion (T2SS/Gsp) and evidence of type VI secretion hallmarks (such as *hcp*, *vgr*G), consistent with export and contact-dependent antagonism. Proteolysis and tissue invasion were indicated by extracellular metalloproteases and serine proteases (e.g., elastase/serralysin-like loci), along with lipases/phospholipases. Adhesion, motility, and biofilm formation were encoded by polar flagellar systems with canonical *flg/flh/fli* structural and export genes, chemotaxis regulators (*che*A/B/R/W/Y/Z), mannose-sensitive hemagglutinin (MSHA) type IV pili, and additional type IV pilus biogenesis modules (*pil*B/M/N/P/Q/T/V/W/Z). Surface and envelope biosynthesis comprised LPS/lipid A and core oligosaccharide pathways (*lpx*A/B/C/D/K/L/M, *waa*A, *kds*A, *rfa*E*/hld*E), alongside capsule/exopolysaccharide contributors such as *gal*U, *ugd*, and nucleotide-sugar interconversion enzymes. Nutrient and iron acquisition determinants included ferrous iron uptake (*feo*A/B), *Ton*B–*Exb*B/*Exb*D–dependent transporters, heme utilization and biosynthesis genes (*hem*A–H), and cytochrome *c* maturation (*ccm*A–I). Redox and stress defenses were reflected by catalase/peroxidase enzymes (*kat*G, *kat*B) and glutathione/cysteine modules (*gsh*A/B, *cys*E/K/M). Cell–cell communication capacity was supported by *lux*S (AI-2 quorum sensing). Multidrug efflux potential was indicated by RND-type pumps (*acr*A*/acr*B) with the outer-membrane channel *tol*C (Appendix A).

### 3.6. Antimicrobial Resistance Genes

Resistome analysis identified multiple antimicrobial resistance determinants in both isolates, consistent with a multidrug-resistant profile, in the *Shewanella* sp. Koi-1 isolate, the class D β-lactamase gene OXA-436 (enzymatic inactivation of β-lactams; decreased susceptibility to carbapenems and penicillins) was detected. The *rsm*A regulator associated with efflux upregulation was present (reduced susceptibility to fluoroquinolones, diaminopyrimidines, and phenicols via increased efflux). A determinant annotated as “*Escherichia coli EF-Tu* mutations conferring resistance to pulvomycin” was also detected (target alteration of elongation factor *Tu*, conferring resistance to elfamycin-class agents) (Appendix A).

In the *A. veronii* Koi-2 isolate, a broader resistance region was observed: sul1 (target replacement with an alternative dihydropteroate synthase, sulfonamide resistance), *qac*EΔ1 (efflux-mediated tolerance to quaternary ammonium compounds and related biocides), *cph*A3 (metallo-β-lactamase, carbapenem hydrolysis), *rsm*A (efflux-associated reduced susceptibility to fluoroquinolones, diaminopyrimidines, and phenicols), *tet*(A) [efflux pump, tetracycline export], *aad*A3 [aminoglycoside adenylyltransferase modifying streptomycin/spectinomycin], and OXA-1157 (class D β-lactamase, hydrolysis of penicillins) (Appendix A).

### 3.7. Bacterial Synergism/Antagonism

Comparative protein clustering with OrthoVenn identified 2076 orthologous gene clusters across the two isolates, of which 1892 were shared. KOI-1 (*Shewanella* sp.) contained 83 isolate-specific clusters, and KOI-2 (*A. veronii*) contained 101 isolate-specific clusters. Total predicted proteins were 4035 (KOI-1) and 4210 (KOI-2). The analysis recovered 1816 single-copy clusters, and 47.01% of all proteins were singletons. Functional annotation placed most genes in metabolism and genetic information processing categories. Successfully annotated genes numbered 2346 (58.1%) in KOI-1 and 2524 (60.0%) in KOI-2. Category counts indicated relatively higher representation of signaling/cellular-process genes in *A. veronii* and of environmental information-processing genes in *Shewanella* sp. antiSMASH detected multiple biosynthetic gene clusters (BGCs) in both genomes (Appendix A). KOI-1 harbored eight BGCs, including RiPP-like, terpene, beta-lactone, NI-siderophore, and arylpolyene clusters; one cluster showed high similarity to an eicosapentaenoic acid (EPA) pathway. KOI-2 contained seven BGCs, comprising terpene, RiPP-like, NRP-metallophore, arylpolyene, and hserlactone clusters; a chromobactin-like NRPS (Type I) cluster was present. KEGG Mapper assigned genes to broad pathway sets, with the highest match counts in global metabolism, secondary-metabolite biosynthesis, microbial metabolism, and cofactor/vitamin biosynthesis. Genes mapped to ABC transporters, two-component systems, amino-acid and carbon metabolism, and to motility/interaction module, including flagellar assembly, bacterial secretion systems, biofilm formation, and quorum sensing (Appendix A).

### 3.8. Environmental Distribution of Shewanella sp. Koi-1 and A. veronii Koi-2

*Shewanella* sp. Koi-1 was detected across 19 ecological categories in the Protologger output, with the highest frequencies in wastewater (18.5%) and freshwater (17.7%), followed by coral (14.7%), insect gut (10.3%), marine water (9.8%), and marine sediment (8.5%). Additional detections were recorded in pig gut (6.9%), activated sludge (6.8%), human vaginal (6.1%), bovine gut (4.9%), chicken gut (4.9%), soil (4.7%), human skin (4.6%), plant (4.2%), rhizosphere (3.1%), human gut (2.3%), mouse gut (2.0%), human lung (1.7%), and human oral (1.1%) (Figure 6).

*A. veronii* Koi-2 was detected across 19 ecological categories in the Protologger output, with the highest frequencies observed in activated sludge (58.8%) and wastewater (43.3%), followed by freshwater (55.5%), rhizosphere (33.2%), plant (16.5%), and insect gut (15.2%) metagenomes. Additional detections were recorded in pig gut (15.2%), soil (13.3%), coral (11.4%), bovine gut (8.1%), chicken gut (7.9%), human skin (7.4%), marine sediment (6.6%), marine water (5.3%), human vaginal (2.2%), mouse gut (2.6%), human lung (2.6%), human gut (3.7%), and human oral (1.0%) metagenomes (Figure 7).

### 3.9. Antimicrobial Susceptibility Results

Antimicrobial susceptibility testing was quality-controlled with *A. salmonicida* subsp. *salmonicida* ATCC 33658; inhibition-zone diameters for ampicillin, gentamicin, florfenicol, oxytetracycline, erythromycin, enrofloxacin, and trimethoprim–sulfamethoxazole fell within the reference ranges specified in CLSI M42-A. Isolates showing no measurable inhibition zone (0 mm) were recorded as resistant.

For *Shewanella* sp. KOI-1, categorical resistance was observed to ampicillin (AMP), amoxicillin (AML), and tetracycline (TE). Species-specific clinical breakpoints are not defined for *Shewanella* in CLSI/EUCAST; therefore, other results are reported descriptively: erythromycin (E) 16 mm, gentamicin (CN) 20 mm, ciprofloxacin (CIP) 21 mm, doxycycline (DO) 16 mm, florfenicol (FFC) 33 mm, and enrofloxacin (ENR) 22 mm.

For *A. veronii* KOI-2, resistance was detected to tetracycline (TE), ampicillin (AMP), amoxicillin (AML), flumequine (UB), trimethoprim–sulfamethoxazole (SXT), and oxytetracycline (OT). EUCAST provides a clinical breakpoint for *Aeromonas* only for ciprofloxacin; the measured CIP 21 mm was interpreted as resistant. For agents without species-specific breakpoints, zone diameters are reported descriptively: erythromycin (E) 16 mm, gentamicin (CN) 20 mm, doxycycline (DO) 16 mm, florfenicol (FFC) 33 mm, and enrofloxacin (ENR) 22 mm.

## 4. Discussion

Rapid environmental change, characterized by rising temperatures, drought, altered hydrology, overharvesting, and expanding polyculture, has been shown to reconfigure aquatic habitats and the microbiotas they sustain [55]. Fish species that typically maintain homeostasis within a familiar microbial community are increasingly forced to coexist with novel assemblages, thereby lowering the threshold for mixed infections. In the context of aquaculture, even minor alterations in water parameters have the potential to significantly alter the composition of the surrounding microbiota, erode the host’s immune system, and create conditions conducive to simultaneous colonization by multiple agents. Co-infections are therefore a routine feature of aquaculture systems, where complex habitats and a diverse pathogen pool foster concurrent infection dynamics [10]. In fish, the routes to co-infection are varied. Inter-microbial interactions may be synergistic or antagonistic, and the host’s condition is pivotal [56]. Parasite burdens and other stressors can depress immune defenses and facilitate secondary bacterial invasion [57]. Despite the growing attention to co-infections, the joint pathogenic potential of *Shewanella* sp. and *A. veronii* in koi has not yet been systematically defined.

We co-isolated *Shewanella* sp. and *A. veronii* from a moribund koi, confirmed identity (culture/biochemistry/MALDI-TOF), and generated draft genomes. Using OrthoVenn3, KEGG (BlastKOALA/Mapper), antiSMASH, and resistome/virulence screens, we performed comparative functional genomics. The integrated in silico analysis delineated complementary and interaction-related pathways (e.g., siderophores, secretion/biofilm, quorum sensing) consistent with potential co-infection.

Mortality events in aquarium fishes are often overlooked because individuals are small and easily replaced. However, diseases arising under long-term aquarium conditions can lead to the loss of valuable broodstock and may also cause emotional distress for owners who develop attachments to their fish. In addition, aquarists commonly manipulate tanks and handle fish and tank water with their bare hands, creating opportunities for pathogen exposure and zoonotic transmission [58]. In the present investigation conducted at a large aquarium wholesaler supplying fish to multiple regions of Turkey, mortality among large koi carp prompted diagnostic evaluation. Affected fish consistently exhibited abdominal distention (commonly referred to by aquarists as “dropsy”), scale loss, lateral (side) swimming, and reduced activity. Necropsy revealed severe abdominal effusion, subcutaneous hemorrhage beneath the scales, and air bladder deformation and displacement. Microbiological analyses identified two dominantly growing bacterial agents, *Shewanella* sp. and *A. veronii*. Similar lesions have been reported in aquarium fish infections in which *Shewanella* spp. and *Aeromonas* spp. are isolated, and co-infection rather than single-pathogen infection is frequently documented in such cases [59,60]. Building on prior research in intensively stocked catfish, where unusual winter mortalities were linked to a mixed bacterial infection and Koch’s postulates were fulfilled for *A. veronii* in the presence of co-isolated *S. putrefaciens*, our findings in koi are consistent with a recurrent *Aeromonas*–*Shewanella* disease ecology [59]. In this particular instance, the co-isolation of *A. veronii* and *Shewanella* sp. from a moribund koi, along with the gross pathology findings (ascites/dropsy, pallor, reduced activity), and the genome-resolved virulence and multidrug-resistance repertoires, align with the clinicopathological picture reported in the catfish outbreak. Taken together, these data indicate that *A. veronii–Shewanella* coinfection is not restricted to a single host species or production system and that its pathogenicity has already been experimentally demonstrated in teleosts. Accordingly, and consistent with the principles of refinement and reduction, a repeat broad experimental infection model may not be necessary to establish causality for the ornamental-fish context. Instead, targeted, hypothesis-driven in vivo work is best limited to verifying specific synergistic mechanisms suggested by our genomic and phenotypic evidence, and to informing mechanism-aware therapy.

Aquaculture remains comparatively nascent from a microbiological standpoint, and the microbial diversity of ornamental aquarium fishes is less thoroughly investigated than that of farmed species. Several studies have reported that many pathogens isolated from aquarium fishes represent previously uncharacterized taxa, and, in some cases, bacterial agents seldom associated with particular hosts have been documented as pathogens [61,62]. In our study, comparative genomics of the *Shewanella* sp. isolate yielded dDDH and ANI values that, when considered together, fell below accepted species-level thresholds (≈95–96% ANI and 70% dDDH), precluding confident identification within any described *Shewanella* species and aligning with reports of taxonomic novelty among aquarium-associated isolates. By contrast, the co-infecting agent, *A. veronii*, highlights the pathogenic potential of this species in aquarium fishes, distinct from more frequently reported *Aeromonas* taxa such as *A. hydrophila*, *A. sobria*, or *A. caviae*. Notably, had identification relied solely on phenotypic or primary assays (MALDI-TOF), these agents might have been assigned to the most commonly encountered *Shewanella* and *Aeromonas* species; whole-genome analysis instead confirmed that both pathogens in this case fall outside the routinely implicated taxa. Collectively, these findings underscore the need to incorporate genome-based identification in diverse infection scenarios and to discriminate rigorously between single-pathogen infections and co-infections.

Mortality and clinical reports indicate that several bacteria commonly detected in ornamental fishes—such as *Vibrio*, *Aeromonas*, and *Plesiomonas*—also cause human infections and therefore exhibit zoonotic potential, as clearly documented by Smith et al. [63]. Consistent with these records, *Shewanella* spp. have been implicated in transmission from fish to humans, including cases associated with fishhook injury–related disease [58,64]. Beyond aquarium husbandry, human exposures in freshwater or marine environments (e.g., fish bites) and in facilities offering “fish pedicure” services have been followed by *Aeromonas*-associated skin and soft-tissue infections, further underscoring the zoonotic character of these agents [65].

Genome-based identification is fundamental to the treatment of animal diseases; without the correct assignment of the etiologic agents, effective therapy cannot be achieved. Given the co-infection identified in this case, antimicrobial susceptibility testing was performed for both pathogens, and treatment protocols were designed to prioritize agents active against both while accounting for known resistance mechanisms in the *Shewanella* sp. Koi-1 isolate, phenotypic resistance to β-lactams such as ampicillin (AMP) and amoxicillin (AML) was concordant with the genomic detection of OXA-436, which encodes a class D β-lactamase responsible for hydrolysis of the β-lactam ring and inactivation of penicillins and selected carbapenems [59]. The observed reduction in susceptibility to fluoroquinolones and phenicols aligned with the presence of *rsmA*, a determinant associated with enhanced efflux activity contributing to multidrug resistance. In addition, the determinant annotated as “*E. coli EF-Tu* mutants conferring resistance to pulvomycin” supports the potential for target-site–mediated resistance to elfamycin-class agents via alterations in elongation factor *Tu*.

*Aeromonas veronii* Koi-2 exhibited broad-spectrum multidrug resistance by both phenotypic and genotypic criteria. Phenotypic resistance to tetracyclines (TE, OT) was concordant with the presence of *tet*(A), which encodes an efflux pump mediating active export of tetracycline. Non-susceptibility to ampicillin and amoxicillin aligned with the detection of OXA-1157 and *cph*A3, β-lactamase determinants associated with hydrolysis of the β-lactam ring. Resistance to trimethoprim–sulfamethoxazole (SXT) correlated with *sul1*, which confers sulfonamide resistance within the folate pathway. In addition, *aad*A3—encoding an aminoglycoside-modifying enzyme—supported resistance to streptomycin/spectinomycin-like agents via enzymatic modification. For both isolates, detection of *rsm*A was consistent with the observed reduced susceptibility to fluoroquinolones and phenicols, which was in line with enhanced efflux activity. Overall, there was a high concordance between phenotypic resistance patterns and genotypic determinants identified via the CARD database, indicating that both isolates employ multiple mechanisms—including enzymatic inactivation, efflux pump overexpression, and target modification—to achieve multidrug resistance.

These findings at the isolate level are consistent with the broader evidence base that positions *Aeromonas*, including *A. veronii*, among the principal bacterial hazards at the aquarium-human interface. There is documented evidence of human infections that are linked to aquaria and aquarium water [66]. Furthermore, *Shewanella* sp. is recognized as a recurrent member of freshwater aquarium microbiota and is implicated in fish disease. The species is flagged among the genera of zoonotic concern. In light of the emerging evidence that numerous freshwater aquarium fish pathogens are zoonoses, and that human cases associated with aquaria encompass *A. veronii* among other enteric/opportunistic agents, our data underscores the imperative for risk-aware husbandry and handling practices, particularly for children, the elderly, and immunocompromised individuals [32,66]. The clinical picture observed in the affected koi—ascites, scale loss, imbalance, and multifocal visceral hemorrhages—align with the combined virulence factors encoded by both isolates. *A. veronii* Koi-2 possessed multiple cytotoxic and enterotoxic components, including aerolysin/hemolysin family toxins and cytotonic enterotoxins, along with secretion systems and motility/biofilm modules that aid in tissue invasion and persistence. Meanwhile, *Shewanella* sp. Koi-1 encoded metalloproteases, complete LPS/LOS biosynthesis pathways, type II secretion systems, and extensive adhesion and biofilm determinants, which support mucosal damage and systemic spread. These features together suggest a plausible mechanism for the severe serositis, ascites, and hemorrhagic lesions observed at necropsy. From a therapeutic perspective, the combined resistome of the two isolates has direct implications for empiric and targeted treatment in ornamental fish medicine. The presence of *bla*OXA-436 in *Shewanella* sp. and *bla*OXA-1157 plus *cph*A3 in *A. veronii*, together with *sul*1, *tet*(A) and *aad*A3, underpins the observed non-susceptibility to penicillins, carbapenems, tetracyclines, sulfonamides and aminoglycosides. These findings suggest that the empirical use of aminopenicillins or tetracyclines in similar cases is not advisable, and instead support the selection of agents with documented in vitro activity against both species, ideally guided by case-specific susceptibility testing and awareness of intrinsic resistance profiles.

From a mechanistic standpoint, our integrated pathway analysis indicates a plausible genomic basis for cooperative resource capture, colonization, and tissue persistence by the *A. veronii* and *Shewanella* co-isolates. It is evident that both genomes encode iron-acquisition systems (e.g., chromobactin-like NRPS in *A. veronii*; NI-siderophore and *Ton*B-dependent receptors in *Shewanella*), secretion and interaction modules (Type II components in *A. veronii*; Hcp-like loci in *Shewanella*), motility/chemotaxis and biofilm formation pathways, quorum-sensing repertoires, and overlapping multidrug determinants (β-lactamases, efflux-associated regulators). Metabolic mapping further highlights complementary capacities, including anaerobic respiration and redox flexibility in *Shewanella*, alongside tissue-damaging enzymes and adhesins in *A. veronii*. AntiSMASH identifies lipid/pigment clusters (such as EPA-linked, arylpolyene) that could enhance oxidative-stress tolerance in mixed communities. Collectively, these computer modeling findings are consistent with the potential for cooperative or at least compatible co-infection dynamics in the field; however, experimental work is still required to confirm any synergistic interactions.

Our co-isolates from koi—*Aeromonas veronii* and *Shewanella* sp.—exhibit genomic features consistent with zoonotic potential. Both genomes encoded over 100 loci associated with virulence, spanning adhesion, secretion, iron acquisition, stress tolerance, and immune evasion modules. These loci indicated capacities relevant to fish disease and compatible with human colonization/infection scenarios. Ecological profiling provides further evidence to support the hypothesis of broad host–habitat connectivity. The genome of *Shewanella* sp. was associated with 19 ecological categories, as determined by the Protologger analysis. The highest representation was observed in wastewater (18.5%) and freshwater (17.7%) samples. Additional signals were detected from human-associated niches, including skin (4.6%), vaginal (6.1%), intestinal (2.3%), lung (1.7%), and oral (1.1%) environments, as well as from animal guts and various environmental reservoirs. Similarly, the genome of *A. veronii* was linked to 19 ecological categories. The highest detection frequencies were recorded in activated sludge (58.8%) and freshwater (55.5%) samples, followed by detections in human skin (7.4%), intestine (3.7%), vaginal (2.2%), lung (2.6%), oral cavity (1.0%), animal guts, and diverse environmental reservoirs. These findings indicate that both *Shewanella* sp. Koi-1 and *A. veronii* Koi-2 isolates are distributed across a wide range of ecological niches, exhibiting a clear predominance in aquatic environments. This pattern underscores their high environmental adaptability and potential involvement in nutrient-rich or anthropogenically influenced aquatic systems. Concurrently, resistome screens identified numerous determinants for β-lactams (including carbapenems), tetracyclines, sulfonamides, aminoglycosides, and efflux-mediated multidrug tolerance, characteristics that could impede clinical management in the event of human transmission.

This work should be viewed as a genome-resolved analysis of a natural outbreak affecting multiple koi, where several fish in the affected aquarium reportedly died with similar clinical and pathological features, as well as microbiological reports. This study specifically focused on a single koi that exhibited the most prominent and representative lesions, allowing for a thorough necropsy and bacterial analysis while minimizing the need for redundant sampling of multiple animals. Instead of trying to reproduce the disease experimentally, this outbreak was used to identify the virulence and resistance profiles of the co-infecting *Aeromonas veronii* and *Shewanella* sp., and to develop mechanistic hypotheses to guide future in vivo studies. This step-by-step approach aligns with the 3R principles (Replacement, Reduction, and Refinement) by utilizing naturally occurring cases, minimizing animal use, and refining future experimental infections to ensure they are both ethically and scientifically justified.

In alignment with a One Health framework, the results obtained provide support for enhanced surveillance that integrates clinical diagnostics in ornamental fish with routine screening of aquarium environments. This is complemented by targeted education for aquarists aimed at mitigating cross-species transmission risks. It is imperative to acknowledge that these inferences are derived from genomic analyses rather than experimental challenge trials and should be validated in vivo.

## 5. Conclusions

In this genome-resolved report, we documented a mixed *Aeromonas veronii*–*Shewanella* sp. co-infection in koi carp associated with severe systemic disease, broad virulence repertoires and concordant multidrug resistance. By integrating culture-based diagnostics with comparative genomics and antimicrobial susceptibility testing, we were able to distinguish between a true co-infection and a single-agent scenario, and to identify resistance determinants with direct consequences for therapy. These findings highlight the value of routine genome-informed diagnostics in ornamental fish medicine, supporting dual-coverage, mechanism-aware treatment while also informing zoonotic risk mitigation at the aquarium–human interface. Future work should extend these observations to larger research series, incorporate histopathology and experimental infection models, and explore targeted prevention strategies along the ornamental fish supply chain.

## Figures and Tables

**Figure 1 microorganisms-14-00036-f001:**
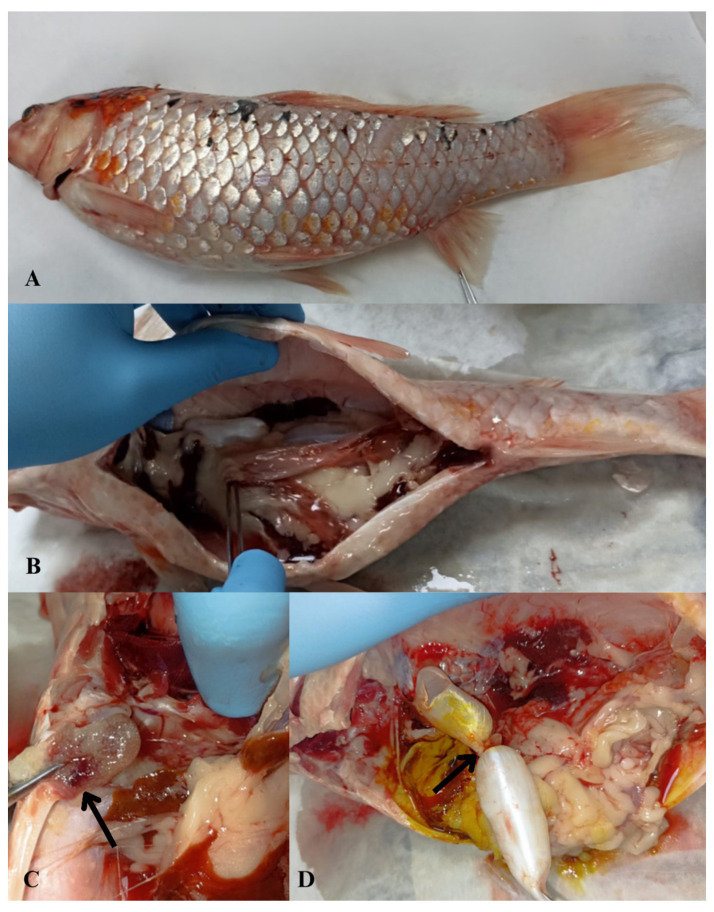
Macroscopic necropsy findings in koi carp (*Cyprinus carpio*) showing abdominal fluid accumulation (ascites) and associated pathological alterations. (**A**) Macroscopic view of abdominal distention due to excessive fluid accumulation within the abdominal cavity; (**B**) abdominal wall distension and hemorrhage observed in visceral organs; (**C**) multifocal hemorrhagic foci in the heart (arrowhead) and visceral organs; (**D**) nodular deformation of the swim bladder (arrowhead).

**Figure 2 microorganisms-14-00036-f002:**
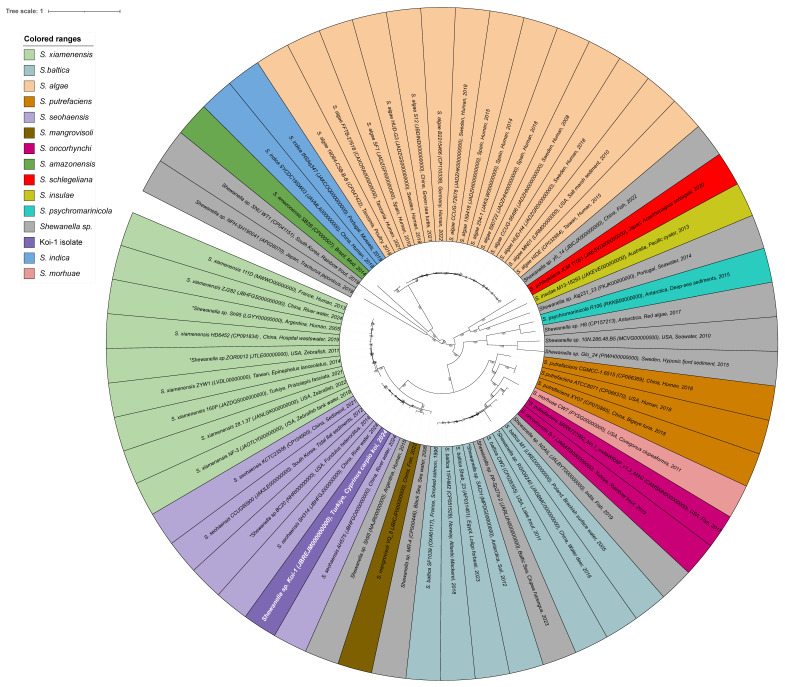
Circular phylogenetic tree of *Shewanella* species generated using the BV-BRC platform. The tree is color-coded by *Shewanella* species, with the Koi-1 isolate highlighted in purple with white lettering. Isolates marked with an asterisk (*) were initially designated as *Shewanella* sp. but were reassigned to specific species following species-identification analyses; these isolates are shown in the color corresponding to their final species assignment.

**Figure 3 microorganisms-14-00036-f003:**
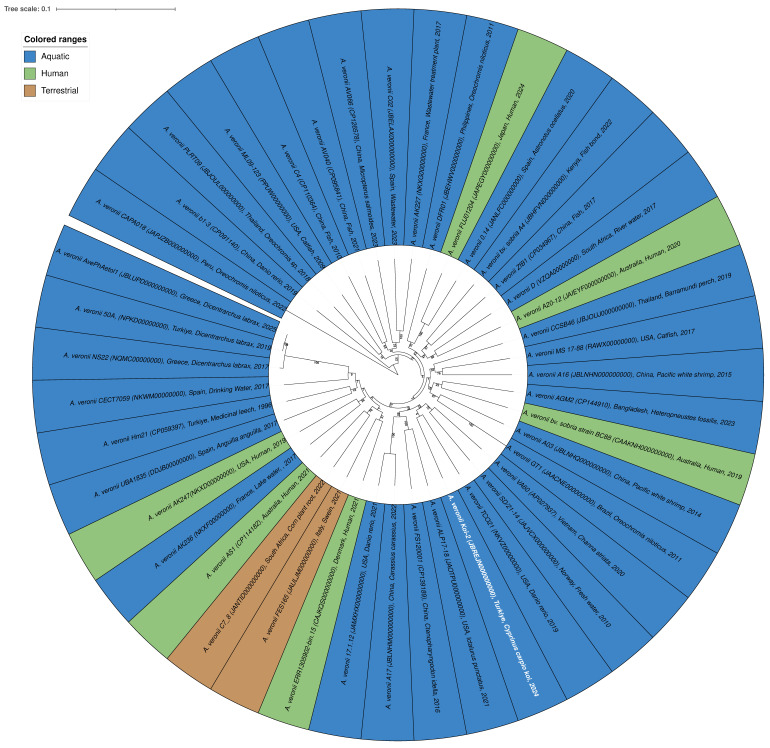
Circular phylogenetic tree of the *A. veronii* species generated using the BV-BRC platform. The tree is color-coded according to the origin of the isolates: blue (aquatic), green (human), and brown (terrestrial). The Koi-2 isolate is highlighted in blue with white text. Phylogenetic analysis indicates that the Koi-2 isolate clusters within the *A. veronii* species complex and shares close phylogenetic relatedness with aquatic-origin isolates.

**Figure 4 microorganisms-14-00036-f004:**
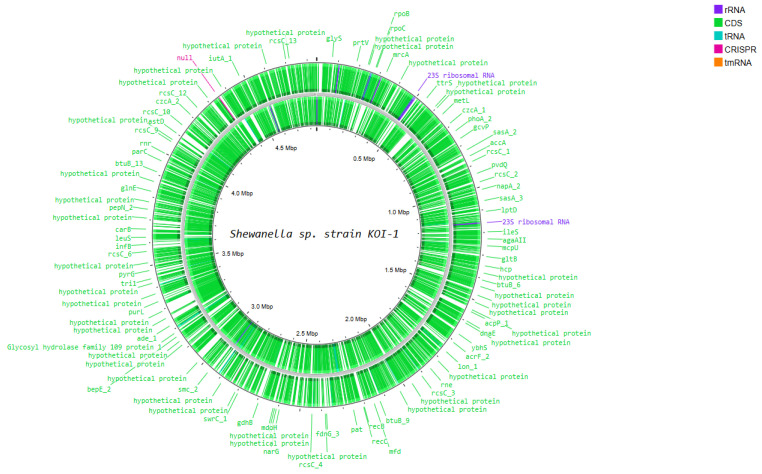
Circular representation of the *Shewanella* sp. Koi-1 genome generated using Proksee, illustrating genome organization and annotated genes.

**Figure 5 microorganisms-14-00036-f005:**
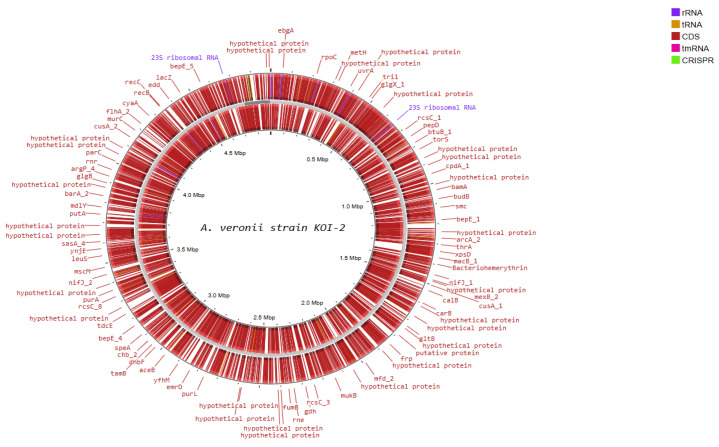
Circular representation of the *A. veronii* Koi-2 genome generated using Proksee, showing genome organization and annotated genes.

**Figure 6 microorganisms-14-00036-f006:**
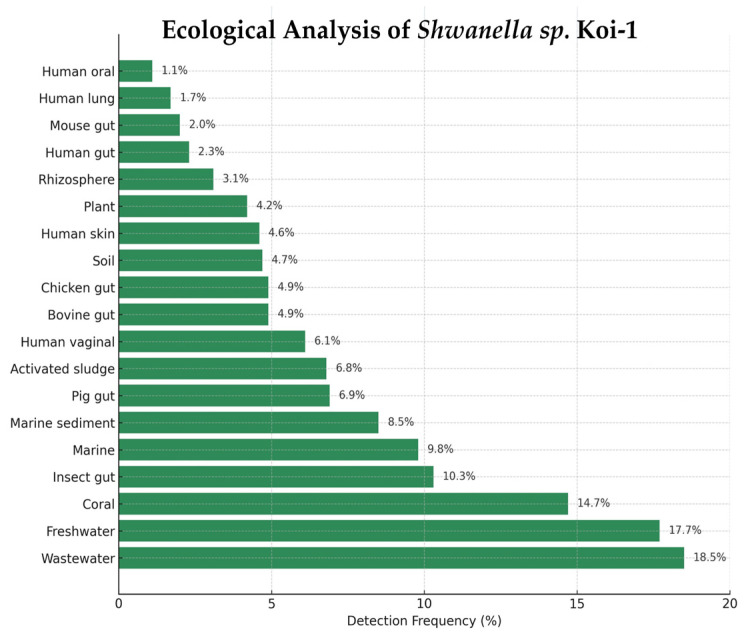
Ecological distribution of *Shewanella* sp. Koi-1 based on Protologger output.

**Figure 7 microorganisms-14-00036-f007:**
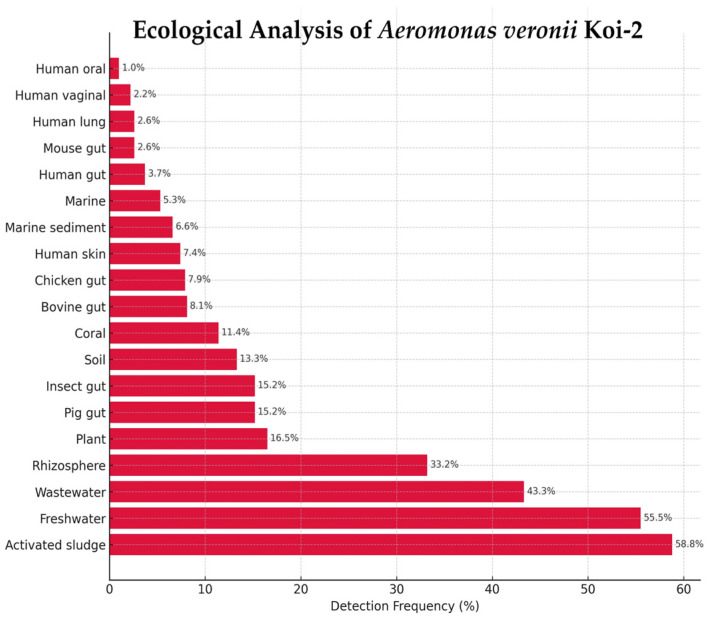
Ecological distribution of *Aeromonas veronii* Koi-2 based on Protologger output.

**Table 1 microorganisms-14-00036-t001:** Predicted prophage regions identified in the genomes of *Shewanella* sp. strain Koi-1 and *A. veronii* strain Koi-2 based on genomic analysis.

Strain	Region	Region Length (Kb)	Completeness	Score	No. of Proteins	Region Position	Most Common Phage	Accession No.	GC (%)
Koi-1	1	45.7	Questionable	70	50	302,8370–3,074,124	PHAGE_Entero_cdtI	NC_009514	47.94
Koi-2	1	38.6	Intact	150	54	1,663,944–1,702,597	PHAGE_Escher_500465_1	NC_049342	58.74
Koi-2	2	38.3	Questionable	80	58	2,001,706–2,040,095	PHAGE_Entero_mEp237	NC_019704	55.76

Note: Each row represents a distinct prophage region predicted within the corresponding genome. Strain Koi-2 harbors two separate prophage regions (Regions 1 and 2) and therefore appears twice in the table.

## Data Availability

The data presented in this study are openly available in [NCBI] at [https://www.ncbi.nlm.nih.gov/, accessed on 12 December 2025].

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
