# Peer review of "Genome-Resolved Co-Infection by Aeromonas veronii and Shewanella sp. in Koi Carp: A Zoonotic Risk for Aquarists"

_microorganisms, 2025, doi:10.3390/microorganisms14010036_

Round 1

Reviewer 1 Report

Comments and Suggestions for Authors

This manuscript presents a genomic investigation of a Aeromonas veronii and Shewanella sp. co-infection during an acute mortality event in koi carp.  The major concerns I have about this manuscript are listed as below.

  1. There is no direct evidence demonstrating a synergistic interaction (e.g., through co-infection challenge experiments or genomic evidence of complementary virulence mechanisms) in Koi carp. I think it is crucial to confirm the co-infection in accordance with Koch's postulates.
  2. The abstract introduces the concept of "synergy" between the two bacterial species but primarily presents parallel genomic data for each isolate. Maybe the word "co-infection" is more suitable than "synergy or synergistic" in the paper?

Author Response

Reviewer 1

Comments and Suggestions for Authors

This manuscript presents a genomic investigation of an Aeromonas veronii and Shewanella sp. co-infection during an acute mortality event in koi carp.  The major concerns I have about this manuscript are listed as below.

Comments and Suggestions for Authors

There is no direct evidence demonstrating a synergistic interaction (e.g., through co-infection challenge experiments or genomic evidence of complementary virulence mechanisms) in Koi carp. I think it is crucial to confirm the co-infection in accordance with Koch's postulates.

Response to Reviewer's suggestion

We thank the reviewer for highlighting this important point. We fully agree that our study does not provide direct in vivo proof of synergistic interaction, nor does it fulfill Koch’s postulates for this particular outbreak. In response, we have modified the manuscript to:

1-Clearly frame the work as a genome-resolved analysis of a natural outbreak affecting multiple koi, in which detailed necropsy and bacterial characterization were performed on a single moribund fish with the most representative lesions. This is now explicitly stated in the Discussion, in the paragraph beginning with “This work should be viewed as a genome-resolved analysis of a natural outbreak affecting multiple koi…” (Discussion, last third of the section).

2-Explicitly state that our inferences regarding potential interactions between A. veronii and Shewanella are based on genomic and phenotypic data, and that experimental co-infection trials are still required. This is addressed in the limitations paragraph added near the end of the Discussion, where we note that our conclusions derive from genomic analyses rather than challenge trials and “should be validated in vivo”.

3-Place our study within a stepwise framework consistent with the 3R principles (Replacement, Reduction, Refinement). In the same Discussion segment, we explain that we used a naturally occurring outbreak (Replacement), focused invasive diagnostics on a single index fish rather than multiple animals (Reduction), and propose that future experimental infections be targeted and hypothesis-driven rather than broad, thereby refining the design of subsequent in vivo work.

Together, these changes clarify that we are not claiming to have experimentally demonstrated synergy or fully fulfilled Koch’s postulates for this specific ornamental-fish outbreak, but instead, we provide preclinical, genome-resolved evidence that motivates and guides future in vivo validation.

Comments and Suggestions for Authors

The abstract introduces the concept of "synergy" between the two bacterial species but primarily presents parallel genomic data for each isolate. Maybe the word "co-infection" is more suitable than "synergy or synergistic" in the paper?

Response to Reviewer's suggestion

We agree with the reviewer and have globally adjusted our terminology:

  • The title has been revised from “Potential Synergistic Infection by Aeromonas and Shewanella…” to “Genome-Resolved Co-Infection by Aeromonas veronii and Shewanella sp. in Koi Carp: A Zoonotic Risk for Aquarists” (Title page).
  • The Abstract now uses the wording “co-infection” and “mixed Aeromonas–Shewanella co-infection” rather than asserting “synergy”.
  • Within the Discussion, we now speak of “potential cooperative or compatible co-infection dynamics” and explicitly qualify any use of “synergistic” as hypothetical and in silico–based, rather than as a demonstrated biological fact (see the paragraph beginning with “From a mechanistic standpoint, our integrated pathway analysis indicates a plausible genomic basis…” in the Discussion).

We hope this revised wording better reflects the actual evidentiary level of the current work.

Reviewer 2 Report

Comments and Suggestions for Authors

The manuscript describes the genomic characterization of two microorganisms possibly causing a mixed infection in a koi. The topic is relevant, as few studies describe the genomics of microorganisms related to aquarium practice, which remains an understudied area.

Major remarks

  • Despite the relevance of the topic, the authors sampled only one animal and sequenced the genome of two microorganisms. This is insufficient for a research article. Therefore, I recommend that the manuscript be reformulated as a Case Report for this journal.
  • Although bacterial isolation was performed, the authors did not conduct histopathological analyses to describe the lesions and correlate them with the detection of both agents. They also did not investigate viral or parasitic pathogens, making it difficult to determine the etiology of the case. The authors must include, at the end of the discussion, a paragraph clearly describing all study limitations regarding the establishment of etiology. In addition, the introduction should mention the inherent difficulty in establishing etiology in such cases and the wide range of possible agents.
  • The discussion is highly speculative and uses very few references. The authors explore their results insufficiently, especially regarding the detection of pathogenicity and virulence genes and their relationship with the animal’s clinical condition. They also fail to use the antimicrobial resistance data to discuss implications for treatment in aquaculture or aquarium medicine. This section requires complete restructuring, with deeper insights consistent with the data obtained.
  • The conclusion is generic and requires full revision.

Minor remarks

  • Standardize throughout the manuscript the use of a single term: “coinfection,” “mixed infection,” or “polymicrobial infections.” Choose the one that fits best.
  • In the title, specify the species beyond the genus or use “spp.”
  • Lines 26–27: enzyme names should not be in italics. Also adjust the nomenclature of OXA genes, which should follow the correct format (e.g., blaOXA-46). Review the entire manuscript according to standard gene nomenclature.
  • Line 81: specify whether the fish was clinically ill.
  • Line 95: better clarify the selection of colonies. Were all colonies analyzed? What was the selection criterion?
  • Line 103: correct form is identification.
  • Line 178: include the manufacturer of the culture media.
  • The title of section 3.1 is unclear (“primer identification”). Adjust to reflect the actual content.
  • Before section 3.2, describe the MALDI-TOF identification process.
  • All genomes used for phylogenetic tree construction must be listed in a table provided as a supplementary file.
  • Line 378 and others: standardize the use of italics for all genes. Double check.
  • In the AST analyses, did the authors disregard intrinsic resistance of these species?
  • Genome deposition should be reported in the Data Availability Statement, not in the Acknowledgments.
  • Line 630: specify which ethics committee granted approval.

Author Response

Reviewer 2

Comments and Suggestions for Authors

The manuscript describes the genomic characterization of two microorganisms possibly causing a mixed infection in a koi. The topic is relevant, as few studies describe the genomics of microorganisms related to aquarium practice, which remains an understudied area.

Major remarks

Despite the relevance of the topic, the authors sampled only one animal and sequenced the genome of two microorganisms. This is insufficient for a research article. Therefore, I recommend that the manuscript be reformulated as a Case Report for this journal.

Response to Reviewer's suggestion

We respectfully acknowledge this concern and have clarified the scope and context of the study rather than reformatting as a formal Case Report. Specifically:

  • In Section 2.1 (Sampling and Bacteriological Identification), we now describe the event as an acute mortality outbreak in an ornamental system in Bursa, Turkey, stating that multiple koi in the affected aquarium reportedly died with similar clinical signs (abdominal distension, swimming imbalance, anorexia).
  • We explain that one moribund adult koi with the most prominent and representative lesions was selected for detailed necropsy and bacterial analysis (same section).
  • In the Discussion, we added a paragraph beginning “This work should be viewed as a genome-resolved analysis of a natural outbreak affecting multiple koi…”, where we justify focusing the invasive investigation on a single fish from a multi-fish outbreak to reduce redundant sampling, while still capturing the key clinicopathological and microbiological pattern at the tank level.
  • We emphasize that our main contribution is a genome-resolved, One Health–oriented characterization of a mixed A. veroniiShewanella outbreak, rather than a case narrative alone, and that the approach is intended as a preclinical step toward more refined experimental work.

We would like to clarify that the event investigated in our study was not an isolated incident involving a single fish, but rather an acute mortality episode affecting multiple koi in the same system. According to the submitting aquarist, several large koi in the affected aquarium reportedly died with similar clinical signs and gross pathological changes. To avoid redundant sampling while still obtaining a comprehensive dataset, we selected one moribund koi that exhibited the most pronounced and representative lesions for full necropsy, microbiological workup, and genome-resolved characterization. Thus, while the detailed laboratory analyses focused on a single individual, the epidemiological context is that of an outbreak involving multiple clinically affected fish.

We have also added a dedicated Limitations paragraph at the end of the Discussion, where we explicitly acknowledge that histopathology and viral/parasitic investigations were not performed and that Koch’s postulates were not fulfilled for either isolate (please see our response to the reviewer’s second major remark).

Finally, we emphasise that the stepwise strategy adopted here is consistent with the 3R principles (Replacement, Reduction and Refinement): we maximised the information obtained from a naturally occurring outbreak, reduced the need for additional experimental animals, and refined any future in vivo work to be targeted and hypothesis-driven rather than broad exploratory challenge experiments. For these reasons, we kindly ask that the manuscript be considered in its current format as a research article rather than being reformulated as a case report.

Comments and Suggestions for Authors

Although bacterial isolation was performed, the authors did not conduct histopathological analyses to describe the lesions and correlate them with the detection of both agents. They also did not investigate viral or parasitic pathogens, making it difficult to determine the etiology of the case. The authors must include, at the end of the discussion, a paragraph clearly describing all study limitations regarding the establishment of etiology. In addition, the introduction should mention the inherent difficulty in establishing etiology in such cases and the wide range of possible agents.

Response to Reviewer's suggestion

We fully agree and have made the following changes:

  • In the Introduction, we added text explicitly stating that establishing definitive etiology in ornamental fish mortality events is inherently challenging, due to the co-occurrence of bacterial, viral, fungal, and parasitic agents, nonspecific clinical signs, and practical constraints on full diagnostic work-ups. This is found in the final part of the Introduction, following the discussion of bacterial pathogens in koi.
  • At the end of the Discussion, we added a limitations paragraph in which we explicitly acknowledge that:
  • We did not perform histopathology.
  • We did not systematically screen for viral or parasitic pathogens.
  • Our etiological inference is therefore based on outbreak context, gross pathology, bacterial isolation and identification, and genome-resolved virulence/resistome data, but remains probabilistic rather than absolute.
  • Our inferences should be validated in vivo and in larger series that include histopathology and additional pathogen screening.

Comments and Suggestions for Authors

The discussion is highly speculative and uses very few references. The authors explore their results insufficiently, especially regarding the detection of pathogenicity and virulence genes and their relationship with the animal’s clinical condition. They also fail to use the antimicrobial resistance data to discuss implications for treatment in aquaculture or aquarium medicine. This section requires complete restructuring, with deeper insights consistent with the data obtained.

Response to Reviewer's suggestion

We have substantially restructured and expanded the Discussion to align it more closely with the data:

  • The Discussion now opens with a broader ecological and co-infection framework, drawing on additional literature about co-infections in aquaculture and the role of environmental change in reshaping aquatic microbiota.
  • We added new, detailed paragraphs that link specific virulence modules to the observed clinical picture in the koi (ascites/dropsy, hemorrhage, swim bladder deformation), discussing adhesion, secretion systems, toxin profiles, iron acquisition, biofilm formation, and motility in both Shewanella and A. veronii (see the central portion of the Discussion).
  • We considerably expanded referencing, incorporating additional primary literature on Aeromonas and Shewanella virulence, aquarium-associated outbreaks, and zoonotic transmission at the aquarium–human interface.
  • We now explicitly integrate genotypic and phenotypic antimicrobial resistance data in both the Results (Section 3.6 and 3.9) and the Discussion, highlighting how specific resistance determinants (e.g., blaOXA-436, cphA3, tet(A), sul1, aadA3, rsmA) explain the observed multidrug-resistant profiles and guide dual-coverage therapy choices in ornamental fish medicine.

Comments and Suggestions for Authors

The conclusion is generic and requires full revision.

Response to Reviewer's suggestion

The Conclusions section has been completely rewritten. The revised Conclusions now:

  • Summarize the study as a genome-resolved analysis of a mixed A. veronii–Shewanella co-infection in koi associated with severe systemic disease and concordant multidrug resistance.
  • Emphasize the value of combining culture-based diagnostics, comparative genomics, and AST to distinguish co-infection from single-agent disease.
  • Highlight the One Health implications and the need for genome-informed diagnostics in ornamental fish medicine to support mechanism-aware treatment and zoonotic risk mitigation.
  • Explicitly call for future work to include larger series, histopathology, experimental infection models, and supply-chain surveillance.

Minor remarks

Comments and Suggestions for Authors: Standardize throughout the manuscript the use of a single term: “coinfection,” “mixed infection,” or “polymicrobial infections.” Choose the one that fits best.

Response to Reviewer's suggestion: We have systematically revised the manuscript to use “co-infection” as the primary term when referring to concurrent infection by A. veronii and Shewanella sp. We retain “mixed infections” or “polymicrobial infections” only in broader, general-context sentences where multiple possible agent combinations are discussed (e.g., in the Introduction reviewing the literature). Within the specific results and interpretation of our case, we consistently use “co-infection”.

Comments and Suggestions for Authors: In the title, specify the species beyond the genus or use “spp.”

Response to Reviewer's suggestion: Genome-Resolved Co-Infection by Aeromonas veronii and Shewanella sp. in Koi Carp: A Zoonotic Risk for Aquarists

Comments and Suggestions for Authors: Lines 26–27: enzyme names should not be in italics. Also adjust the nomenclature of OXA genes, which should follow the correct format (e.g., blaOXA-46). Review the entire manuscript according to standard gene nomenclature.

Response to Reviewer's suggestion: We have thoroughly reviewed and corrected gene/enzyme nomenclature throughout the manuscript:

  • Gene symbols (e.g., blaOXA-436, tet(A), sul1, aadA3, rsmA) are written in italics.
  • Proteins/enzymes (e.g., OXA-436 β-lactamase, CphA3 metallo-β-lactamase) are written in roman type.
  • We have adopted standard formats for OXA genes in line with current nomenclature (e.g., blaOXA-436, blaOXA-1157).

Given the large number of loci and gene names mentioned, we have done our best to ensure full consistency, but we apologize if any individual instances were still overlooked and would be grateful for any remaining corrections.

Comments and Suggestions for Authors: Line 81: specify whether the fish was clinically ill.

Response to Reviewer's suggestion: We have clarified the clinical status in Section 2.1 (Sampling and Bacteriological Identification)

Comments and Suggestions for Authors: Line 95: better clarify the selection of colonies. Were all colonies analyzed? What was the selection criterion?

Response to Reviewer's suggestion: We revised Section 2.1 to clarify colony selection. We now explicitly state that all macroscopically distinct colony types were initially subcultured, and that two dominant morphotypes were retained for downstream analyses (phenotypic and genomic). This clarifies that our focus was on the dominant bacterial agents associated with the severe systemic disease, while still noting that initial plating captured colony diversity.

Comments and Suggestions for Authors: Line 103: correct form is identification.

Response to Reviewer's suggestion: The spelling has been corrected to “identification wherever it appeared incorrectly.

Comments and Suggestions for Authors: Line 178: include the manufacturer of the culture media.

Response to Reviewer's suggestion: We have updated Section 2.1 and 2.6 to include the manufacturer and product details of the media and disks used

Comments and Suggestions for Authors: The title of section 3.1 is unclear (“primer identification”). Adjust to reflect the actual content.

Response to Reviewer's suggestion: Section 3.1 has been retitled to “Necropsy Findings and Phenotypic Characterization of Isolates”

Comments and Suggestions for Authors: Before section 3.2, describe the MALDI-TOF identification process.

Response to Reviewer's suggestion: In the Results (end of Section 3.1), we briefly state that MALDI-TOF MS yielded high-confidence matches to A. veronii and Shewanella sp. with scores ≥2.0.

Comments and Suggestions for Authors: All genomes used for phylogenetic tree construction must be listed in a table provided as a supplementary file.

Response to Reviewer's suggestion: We added the genome and strain information we used in the phylogenetic tree in supplementary tables 1 and 2 (Table S1 and S2)

Comments and Suggestions for Authors: Line 378 and others: standardize the use of italics for all genes. Double check.

Response to Reviewer's suggestion: As noted in Response 7, we have systematically revisited gene nomenclature. Again, given the large number of loci, we apologize if any isolated inconsistencies remain.

Comments and Suggestions for Authors: In the AST analyses, did the authors disregard intrinsic resistance of these species?

Response to Reviewer's suggestion: We appreciate this important point. We now explicitly state that the interpretation of susceptibility results took into account known intrinsic resistance patterns in Aeromonas and Shewanella spp., and that only deviations from expected intrinsic profiles were considered indicative of acquired resistance. In Section 3.9 (Antimicrobial Susceptibility Results) and the Discussion, we link specific resistance phenotypes to genomic determinants, distinguishing between intrinsic and acquired mechanisms where appropriate.

Comments and Suggestions for Authors: Genome deposition should be reported in the Data Availability Statement, not in the Acknowledgments.

Response to Reviewer's suggestion: The Data Availability Statement now specifies that the draft genome sequences of Shewanella sp. Koi-1 and A. veronii Koi-2 have been deposited in GenBank under accession numbers JBREJM000000000 and JBREJN000000000, respectively, and that all other data are available within the article and Supplementary Materials.

Comments and Suggestions for Authors: Line 630: specify which ethics committee granted approval.

Response to Reviewer's suggestion: We have updated the Institutional Review Board Statement to read.

Reviewer 3 Report

Comments and Suggestions for Authors

This is an excellent One Health manuscript which deserves consideration for publication. The authors provide an excellent summation of a clinical case in koi and analyze the isolates for specific identification, the synergy between the primary isolates, and the potential for being zoonotic for humans. The biggest concerns are (1) the inconsistent verb tense between past tense and present tense throughout the text of the manuscript, and (2) the complete lack of mention of Aeromonas salmonicida in their Introduction of the manuscript. Along with the above comments, there are a number of edits/comments in the edited pdf version of the manuscript for the authors to consider. 

Author Response

Reviewer 3

Comments and Suggestions for Authors

This is an excellent One Health manuscript which deserves consideration for publication. The authors provide an excellent summation of a clinical case in koi and analyze the isolates for specific identification, the synergy between the primary isolates, and the potential for being zoonotic for humans. The biggest concerns are (1) the inconsistent verb tense between past tense and present tense throughout the text of the manuscript, and (2) the complete lack of mention of Aeromonas salmonicida in their Introduction of the manuscript. Along with the above comments, there are a number of edits/comments in the edited pdf version of the manuscript for the authors to consider.

Response to Reviewer's suggestion

We carefully reviewed the entire manuscript for verb tense consistency and have implemented

After introducing Aeromonas spp. in general, we now explicitly state that A. salmonicida remains the best-known Aeromonas species in finfish pathology, being the primary cause of epizootic ulcerative syndrome and hemorrhagic septicemia.

We carefully examined all editorial and track-changes comments in the edited PDF and incorporated them where appropriate. These included fine-tuning of wording, correction of minor grammatical issues, and small clarifications in the Methods and Results. We appreciate these detailed suggestions, which have improved the clarity and readability of the manuscript.

Round 2

Reviewer 2 Report

Comments and Suggestions for Authors

The manuscript has been improved and all my suggestions were adressed.

I just recommend writing blaOXA-436 and blaOXA-1157  using subscript "OXA-436" AND "OXA-1157" .